# Significance of Plasma Irisin, Adiponectin, and Retinol Binding Protein-4 Levels as Biomarkers for Obstructive Sleep Apnea Syndrome Severity

**DOI:** 10.3390/biom13101440

**Published:** 2023-09-24

**Authors:** Nevin Fazlıoğlu, Pelin Uysal, Sinem Durmus, Sibel Yurt, Remise Gelisgen, Hafize Uzun

**Affiliations:** 1Department of Pulmonary Medicine, Namık Kemal University, 59010 Tekirdag, Turkey; nevinfazlioglu@gmail.com; 2Maslak Hospital, Faculty of Medicine, Department of Pulmonary Medicine, Acibadem Mehmet Ali Aydinlar University, 34752 Istanbul, Turkey; drpelinuysal@gmail.com; 3Department of Biochemistry, School of Medicine, Istanbul University-Cerrahpasa, 34320 Istanbul, Turkey; durmus.sinem@gmail.com (S.D.); remise.gelisgen@iuc.edu.tr (R.G.); 4Basaksehir Cam and Sakura State Hospital, Department of Pulmonary Medicine, University of Health Sciences, 34480 Istanbul, Turkey; yurtsibell@gmail.com; 5Department of Biochemistry, Faculty of Medicine, İstanbul Atlas University, 34403 Istanbul, Turkey

**Keywords:** obstructive sleep apnea, irisin, retinol-binding protein-4, adiponectin, apnea hypopnea index

## Abstract

Objective: Obstructive sleep apnea syndrome (OSAS) is a common sleep disorder that is caused by the reduction or cessation of airflow in the upper airway. Irisin, retinol-binding protein-4 (RBP-4), and adiponectin are the three significant factors in the metabolic process of the human body. The objective of this study was to investigate whether plasma irisin, RBP-4, and adiponectin levels are associated with the severity of OSAS. Methods: According to inclusion and exclusion criteria, 125 patients with OSAS and 46 healthy, gender-matched controls were included in this study. The patients were classified according to the apnea hypopnea index (AHI) as 14 mild cases (5 < AHI < 15), 23 moderate OSAS cases (15 < AHI < 30), and 88 severe OSAS cases (AHI > 30). The plasma irisin, RBP-4, and adiponectin levels were measured and compared between groups. Results: RBP-4 levels were higher in severe OSAS compared to other groups, and irisin levels were significantly lower in severe OSAS compared to other groups. There was a negative correlation between irisin and RBP-4 (r = −0.421; *p* < 0.001), and irisin and AHI (r = −0.834; *p* < 0.001), and a positive correlation between irisin and adiponectin (r = 0.240; *p* = 0.002). There was a negative correlation between RBP-4 and adiponectin (r = −0.507; *p* < 0.001) and a positive correlation between RBP-4 and AHI (r = 0.473; *p* < 0.001). As a predictor of OSAS, adiponectin showed the highest specificity (84.8%) and RBP-4 the highest sensitivity (92.0%). Conclusion: Circulating adiponectin, irisin, and RBP-4 may be new biomarkers in OSAS patients in addition to risk factors such as diabetes, obesity, and hypertension. When polysomnography is not available, these parameters and clinical data can be used to diagnose the disease. As a result, patients with an AHI score greater than thirty should be closely monitored for metabolic abnormalities.

## 1. Introduction

Obstructive sleep apnea syndrome (OSAS) is a common sleep disorder that is caused by the reduction or cessation of airflow in the upper airway. The symptoms of OSAS are presented by a significant fragmentation of sleep, the existence of hypoxemia and hypercapnia, increased daytime sleepiness, as well as a marked disruption of functionalities in numerous aspects of daily life [1]. The prevalence of OSAS is estimated at around 6–13% of among Western countries [2]. The hallmark symptom of OSAS is daytime sleepiness, which reduces the quality of life because of disturbed sleep. There are numerous other symptoms, such as depression, headaches, memory problems, and concentration problems. Furthermore, OSAS is strongly linked to cardiovascular events such as hypertension (HT), rhythm problems, myocardial infarctions, stroke, and sudden death [1].

Irisin is produced during exercise from myokines and released into the body’s circulation. The recent consideration of irisin suggests that it is also produced in adipose tissue and is also accepted as an adipokine [3,4]. Irisin is reported to reverse diet-associated obesity by promoting the adipocyte-like cells [4]. It is also well established that serum irisin level correlates negatively with body mass index (BMI) and the level of irisin is reported to be lower in individuals with obesity compared with lean individuals [5]. Although irisin is a critical factor in obesity, there are restricted numbers of studies that have investigated irisin in patients with OSAS [3,4]. BMI was the main contributor to circulating irisin after controlling for age, waist-to-hip ratio, and insulin sensitivity [3]. Serum irisin concentrations were negatively correlated with the presence and severity of OSAS. Irisin is also considered to have a place in the pathophysiology of OSAS depending on its anti-inflammation effects [4].

Retinol-binding protein-4 (RBP-4) is an adipokine that has been shown to affect glucose metabolism and regulate insulin resistance in peripheral tissues [6]. It has been reported that the increase in RBP-4 levels was closely associated with obesity and impaired glucose tolerance. Moreover, the level of RBP-4 was shown to decrease after regular exercise and the weight loss associated with bariatric surgery [7]. As a result, RBP-4 can be used to predict diabetes risk as well as cardiovascular events. Regarding OSAS, it can be said that the number of studies is limited in terms of exploring the association between OSAS and plasma RBP-4. Two previous studies have mentioned this issue and have reported no significant correlations between apnea-related parameters and RBP-4 [8,9].

Adiponectin is a 30 kDa fat protein hormone that is produced in adipose tissue and secreted into the systemic circulation [10]. Adiponectin was shown to have benefits for metabolic parameters as well as the cardiovascular system. It has also been shown that adiponectin has anti-inflammatory effects [11]. Thus, adiponectin is currently considered to have protective effects against obesity-related outcomes [12]. Several studies indicated no difference in plasma adiponectin level in patients with OSAS; however, some of them reported decreased levels of adiponectin in patients with OSAS compared with individuals without OSAS [13,14,15,16].

This study investigated whether plasma irisin, RBP-4, and adiponectin levels are associated with the severity of OSAS in patients with obesity and type 2 diabetes mellitus (type 2 DM).

## 2. Materials and Methods

### 2.1. Informed Consent

The protocol for sample collection was approved by Acibadem Mehmet Ali Aydinlar University, Medical Faculty Ethics Committee (2016-8/32) and was carried out according to the requirements of the Declaration of Helsinki. All patients were fully informed of the study procedures before they gave their consent.

### 2.2. Study Population

We conducted our single-center and prospective study between June 2016 and May 2017 on participants who applied to our sleep laboratory for the first time and underwent sleep testing. According to the American Academy of Sleep Medicine (AASM) guidelines, 14 mild cases (5 < AHI < 15), 23 moderate OSAS cases (15 < AHI < 30), and 88 severe OSAS cases (AHI > 30) were included.

The control group consisted of 46 healthy individuals who were admitted to our sleep laboratory with suspicion of OSAS and confirmed to not have OSAS through polysomnography (PSG; AHI < 5 h). Controls were matched according to gender. They were subjected to the same exclusion criteria as patients with OSAS.

### 2.3. Exclusion Criteria

Patients previously diagnosed with OSAS and using continuous positive airway pressure (CPAP) therapy were excluded from this study. We excluded patients with known cancer; chronic inflammatory disease; any systemic infection; a known acute coronary syndrome; valvular heart disease; and thyroid, renal, or hepatic dysfunction, as well as patients taking glucocorticoids or nonsteroidal anti-inflammatory drugs.

### 2.4. Polysomnography (PSG)

All participants underwent PSG (Embla N 700 sleep system; Natus Medical Incorporated, Pleasanton, CA, USA) testing. At least 6 h of PSG data was recorded. PSG recordings included 6-channel electroencephalography, 2-channel electrooculography, 2-channel submental electromyography, oxygen saturation via an oximeter finger probe, respiratory movements via chest and abdominal belts, airflow via both nasal pressure sensor and oronasal thermistor, electrocardiography, and leg movements via both tibial anterolateral electrodes.

Sleep stages were scored in 30 s periods by a registered sleep physician certified according to AASM criteria. Scoring as apnea was based on a ≥90% decrease in the respiratory signal (obtained with an oronasal thermal sensor in the diagnostic test) during sleep compared to baseline and a ≥90% signal loss lasting ≥10 s. In order to be scored as hypopnea, the respiratory signal during sleep (obtained with a nasal cannula in the diagnostic test) decreased by ≥30% compared to the baseline value, the ≥30% signal loss lasted for ≥10 s, and the pre-event basal oxygen saturation decreased by ≥3% or the event ended with arousal. The number of apneas and hypopneas per hour of sleep was calculated to obtain AHI. OSAS severity was evaluated as mild, moderate, and severe according to AHI values of 5–14, 15–29, and >30, respectively.

The degree of obesity is measured using body mass index (BMI). BMI is calculated using weight in kilograms divided by height in meters squared. For adults, a BMI of 25.0–29.9 kg/m^2^ is defined as overweight and ≥30 kg/m^2^ as obese [17].

### 2.5. Sample Collection and Measurements

Blood samples were collected in EDTA-containing tubes and anticoagulant-free tubes in the morning after 12–14 h of fasting. After centrifugation at 2500× *g* for 5 min, the plasma and serum separated at least in 30 min. Each sample was separated into four aliquots and samples were stored at −80 °C until biochemical analysis.

### 2.6. Measurement of Plasma Irisin Levels

Plasma irisin levels were measured through the competitive enzyme-linked immunosorbent assay (ELISA) method using commercially available kit (Irisin ELISA, Biovendor, Cat. No: RAG018R, Brno, Czech Republic ), according to the manufacturer’s directions. The coefficients of intra- and inter-assay variations were 4.2% (*n* = 20) and 4.9% (*n* = 20), respectively.

### 2.7. Measurements of the Plasma RBP-4 Levels

Plasma RBP-4 levels were measured using a sandwich ELISA kit (Human RBP4 Immunoassay, Quantikine^®^ ELISA, Cat. No. DRB400, USA), according to the manufacturer’s directions. The coefficients of intra- and inter-assay variations were 4.3% (*n* = 20) and 5.4% (*n* = 20), respectively.

### 2.8. Measurement of Plasma Adiponectin Levels

Plasma adiponectin levels were measured through sandwich ELISA method using commercially available kit (Human Adiponectin ELISA Kit, Assaypro LLC, Cat. No. EA2500-1, USA), according to the manufacturer’s directions. The coefficients of intra- and inter-assay variations were 4.5% (*n* = 20) and 5.4% (*n* = 20), respectively.

### 2.9. Statistical Analysis

Histogram, q-q plots, and Shapiro–Wilk’s test were applied to assess the data normality. Levene’s test was used to test the homogeneity of variances. To compare the differences among AHI groups, one-way analysis of variance (ANOVA) and Kruskal–Wallis H tests were performed for quantitative variables, while chi-square analysis was performed for qualitative variables. Tukey’s HSD and Siegel–Castellan tests were applied for multiple comparisons. Pearson correlation coefficients were found to determine the existence of an accepted relationship and the magnitude and direction of the relationships between the variables. To evaluate the relationship between plasma adiponectin, irisin, and RBP-4 levels and the development of OSAS, cut-off values for these parameters were determined through receiver operating characteristic (ROC) analysis. All analyses were conducted using TURCOSA (Turcosa Analytics Ltd. Co. Turkey, http://www.turcosa.com.tr) and JAMOVI 2.3.18 statistical softwares (accessed on 30 April 2023). A *p* value less than 5% was considered statistically significant.

## 3. Results

### 3.1. Comparison of General Characteristics

Height (*p* = 0.228), gender (*p* = 0.431), obesity (*p* = 0.137), and mean oxygen saturation (*p* = 0.070) did not statistically differ between groups. A significant difference was found between the mean ages of AHI groups (*p* < 0.001). The difference in age was due to the difference between AHI < 5 and AHI > 30 groups. The lowest age (41.67 ± 11.04) was observed in the control group with an AHI value less than 5, while the highest mean age (52.57 ± 10.62) was observed in the group with an AHI value greater than 30. A statistically significant difference was found between the mean weights of the AHI groups (*p* < 0.001). The control group with an AHI of less than 5 had the lowest mean weight (74.61 ± 13.91), while the group with an AHI of greater than 30 had the highest mean weight (93.55 ± 16.37). A statistically significant difference was also found between the mean body mass indexes of the AHI groups (*p* < 0.001). This difference in body mass index was due to the AHI < 5 group being different from the 15 < AHI < 30 and AHI > 30 groups on average. The lowest mean body mass index (25.66 ± 3.59) was observed in the AHI < 5 group, while the highest mean body mass index (33.34 ± 6.52) was observed in the group with an AHI value greater than 30. While 100% of the AHI < 5 and 5 < AHI < 15 groups did not have DM, 91.3% of the 15 < AHI < 30 group and 78.4% of the AHI > 30 group did not have DM. There was a significant relationship between AHI groups and the HT variable (*p* < 0.001). While 100% of the AHI < 5 group had no HT, 76.9% of the 5 < AHI < 15 group, 47.8% of the 15 < AHI < 30 group, and 56.3% of the AHI > 30 group had no HT (Table 1).

### 3.2. Comparisons of Plasma Irisin, RBP-4, and Adiponectin Levels between Groups

A statistically significant difference was found between the mean irisin of AHI groups (*p* < 0.001). The mean irisin of the severe OSAS group was statistically significantly lower than those of the other groups. A statistically significant difference was found between the mean adiponectin levels of AHI groups (*p* < 0.001). The median adiponectin levels of controls were significantly higher compared to other OSAS groups. However, there was no statistically significant difference in adiponectin levels between patient groups. The plasma RBP-4 level was found to be significantly different between groups. RBP-4 levels were higher in severe OSAS compared to the other groups. Also, RBP-4 levels were higher in moderate OSAS compared to the control group (Table 1).

The results of ANCOVA applied to control the confounding factors between the factors are shown in Table 2. Accordingly, there was a significant difference between AHI groups for irisin, but age, BMI, DM, and HT were not significant and no interaction was observed. For RPB-4, a significant difference was found for AHI groups and BMI, but other factors and interactions were not significant. For adiponectin, BMI was the only significant factor, there were no difference between AHI groups, and there were no interactions between factors.

### 3.3. Comparisons of Plasma Irisin, RBP-4, and Adiponectin Levels between Comorbidity in Cases

Table 3 shows irisin, RBP-4, and adiponectin levels according to comorbidity groups. Irisin and adiponectin levels were significantly lower and RBP-4 levels were significantly higher in patients with obesity compared to those without obesity. Irisin and adiponectin levels were significantly lower and RBP-4 levels were significantly higher in patients with DM compared to those without DM. RBP-4 levels were significantly higher and adiponectin levels were significantly lower in patients with HT compared to those without HT, whereas there was no significant difference between irisin levels.

### 3.4. Correlation Analysis

Table 4 shows the correlation coefficients between the variables. The correlation coefficient between irisin and RBP-4 variables is statistically significant and shows a moderate negative relationship (r = −0.421, *p* < 0.001) (Figure 1). The correlation coefficient between irisin and adiponectin variables is statistically significant and shows a weak positive relationship (r = 0.240, *p* = 0.002) (Figure 2). The correlation coefficient between irisin and AHI variables shows a statistically significant negative strong relationship (r = −0.834, *p* < 0.001). The correlation coefficient between irisin and BMI variables shows a statistically significant negative weak relationship (r = −0.249, *p* = 0.001). The correlation coefficient between RBP-4 and adiponectin variables shows a statistically significant negative and moderate relationship (r = −0.507, *p* < 0.001) (Figure 3). The correlation coefficient between RBP-4 and AHI variables shows a statistically significant positive moderate relationship (r = 0.473, *p* < 0.001). The correlation coefficient between RBP-4 and BMI variables shows a statistically significant positive and moderate relationship (r = 0.535, *p* < 0.001). The correlation coefficient between adiponectin and AHI variables does not show a statistically significant relationship (r = −0.118, *p* = 0.190). The correlation coefficient between adiponectin and BMI variables shows a statistically significant negative strong relationship (r = −0.768, *p* < 0.001). There is a positive correlation between AHI and BMI (r = 0.205, *p* = 0.022) in the patient group (Table 4).

### 3.5. ROC and Regression Analysis

Table 5 shows the evaluation of the ROC analysis of adiponectin, RBP-4, and irisin levels as predictors of OSAS. RBP-4 (AUC:0.893; 0.837–0.935) and adiponectin (AUC:0.826; 0.761–0.879) have a good AUC value, and irisin (AUC:0.690; 0.615–0.758) has a satisfactory AUC value. RBP-4 has a sensitivity of 92.0% and a specificity of 67.3% with a cut-off of 32.34 for OSAS. Adiponectin has a sensitivity of 74.4% and a specificity of 84.8% with a cut-off of 7.42 for OSAS.

## 4. Discussion

In the current study, we were able to show that plasma irisin and RBP-4 levels differed between OSAS groups classified according to AHI. The plasma adiponectin levels were found to be significantly higher in the control group; however, the adiponectin levels in the OSAS groups were comparable. As a predictor of OSAS, adiponectin showed the highest specificity (84.8%) and RBP-4 the highest sensitivity (92.0%). Furthermore, significant correlations were discovered between irisin, adiponectin, and RBP-4, as well as AHI and BMI. According to ANCOVA results, irisin was different between AHI groups, not affected by other factors. For RBP-4, there was a difference between AHI groups, also related to BMI, but there was no interaction between them. The situation was different for adiponectin; when we performed ANCOVA, there was no difference between AHI groups for adiponectin. The only important factor affecting adiponectin was BMI.

Irisin is a myokine and its secretion is considered to be associated with exercise. Because it is also secreted from adipose tissue, it is also accepted as an adipokine [16,18]. Irisin was reported to increase energy without food intake in an experimental model [19]. It was also reported that irisin could reverse obesity via effecting adipose cells [20]. Human studies also showed that irisin levels correlated negatively with BMI and its level was found to be lower in obese individuals compared with non-obese ones [21]. Regarding the association between irisin and OSAS, there are two previous studies that mentioned this issue. In a previous study, serum irisin concentration was found to be significantly lower in OSAS patients compared to the control group. It was also shown that the serum irisin level decreased more in patients with severe OSAS compared with those with mild and moderate OSAS. A significant negative correlation was found between serum irisin level and OSAS severity [4]. The second study found no significant difference in serum irisin levels between the mild-to-moderate OSAS group and the severe OSAS group. The authors concluded that the irisin–BDNF axis affected daytime sleepiness [22]. The current study’s findings support the literature in terms of a lower irisin level in severe OSAS compared to mild-type. Furthermore, we discovered that plasma irisin levels were significantly and negatively correlated with plasma RBP-4 levels while being positively correlated with plasma adiponectin levels. The results of the correlation analysis are the first in the literature and show the relationship between plasma irisin, RBP-4, and adiponectin in OSAS patients with obesity and type 2 DM.

Adiponectin has a key role in insulin resistance and binds to the adipose tissue of numerous organ systems. There are several studies that investigated the adiponectin levels in patients who suffered from OSAS. In general, serum adiponectin levels were reported to be lower in OSAS patients compared with healthy subjects [13,23,24,25]. Additionally, the severity of OSAS, which was determined using AHI, was reported to be associated with decreased serum levels of adiponectin [24]. A recent meta-analysis reported significantly decreased plasma/serum levels of adiponectin in OSAS patients and it was concluded that adiponectin had a potential role in the pathogenesis of OSAS [16]. In the present study, we found that the control group had significantly higher serum adiponectin levels compared with other groups. Furthermore, significant correlations were discovered between serum adiponectin levels and serum irisin and RBP-4 levels, as well as BMI. Thus, our results confirmed the previous data and showed novel findings of a correlation between serum adiponectin, irisin, and RBP-4 in OSAS patients with obesity and type 2 DM.

RBP-4 is primarily produced in the liver and adipose tissue and is secreted into the circulation. RBP4 is a transporter that moves retinol from the liver to peripheral tissues for the production of retinoic acid (RA). Besides its transporter duty, RBP4 causes the secretion of proinflammatory cytokines and is responsible for the activation of antigen-presenting cells in adipose tissue [26,27]. There has been strong evidence that shows that high levels of RBP4 have significant roles in the development of metabolic diseases and the induction of oxidative stress and inflammation [17,28,29,30]. There have been limited data sets that have investigated the role of RBP4 in OSAS patients. Makino and his colleagues reported that plasma RBP4 levels in moderate-to-severe OSAS patients were higher than in control subjects. They also concluded that visceral obesity was associated with higher levels of RBP4 in OSAS patients. [9]. Nena and her coworkers reported that serum RBP-4 levels were not associated with OSAS-related parameters and demonstrated that serum RBP-4 levels decreased under continuous positive airway pressure treatment [8]. In present study, we demonstrated that plasma RBP-4 levels were significantly higher in the 15 < AHI < 30 group and AHİ > 30 group compared with other groups and supports the literature. As a predictor of OSAS, adiponectin showed the highest specificity (84.8%) and RBP-4 the highest sensitivity (92.0%). Furthermore, RBP-4 levels were found to be significantly and negatively correlated with plasma irisin and adiponectin levels, and this is the novel finding of our study.

Fasting serum RBP-4, adiponectin, and leptin were associated with peripheral insulin sensitivity, were abnormal in the first-degree relatives of type 2DM parents, and correlated with the soleus intramyocellular lipid content and with the intrahepatic lipid content. Serum RBP-4 was a robust marker of insulin resistance. Serum RBP-4 and adiponectin concentrations reflected ectopic fat accumulation in humans [31]. Tabak et al. [32] reported that irisin, RBP-4, and adiponectin are hallmarks of metabolic syndrome (MetS), which is related to low-grade inflammation. Irisin and adiponectin might contribute to the development of MetS and may also represent novel MetS components [33]. Adiponectin is significantly negatively correlated with irisin in chronic obstructive pulmonary disease (COPD) [33]. In the current study, there would be significant associations between plasma irisin, adiponectin, and RBP-4 in OSAS, but the precise mechanisms are still unclear.

The gold standard for diagnosis then patients with suspected OSAS is PSG performed in a sleep laboratory [34]. AHI is the most commonly used criterion for the diagnosis and severity of OSAS as a result of polysomnographic study. In the current study, the correlation coefficient between RBP-4 and AHI variables showed a statistically significant positive moderate relationship. The correlation coefficient between adiponectin and AHI variables did not show a statistically significant relationship. There was a positive correlation between AHI and BMI in the patient group. Although PSG has been accepted as the gold standard in the diagnosis of OSAS since 1965, it is expensive, time-consuming, and requires special teams and devices [35]. Due to the relationship between AHI and circulating adiponectin, irisin, and RBP-4 levels, they may be preferred in the future because they are cheap, easy, and short-term in cases where PSG is not performed. 

In current study, biomarkers were assessed in patients with OSAS-related comorbidities such as DM, obesity, and HT. Irisin and adiponectin levels decreased and RBP-4 levels increased in patients with obesity and DM. RBP-4 levels were higher and adiponectin levels were lower in patients with HT. Indeed, the negative correlation between AHI, which reflects the severity of the disease, with adiponectin and irisin, and the positive correlation with RBP-4 may indicate potential new biomarkers for determining the risk levels in OSAS patients. It would be possible to use these molecules for risk determination or even diagnosis as our knowledge increases on their roles in diseases with comorbidity and certain physiological cases, as well as their impact mechanisms.

The present study has several limitations. Firstly, we did not evaluate other metabolic parameters such as serum lipid levels. This issue is considered a limitation. Secondly, the sample sizes of groups can be considered too small for drawing a general conclusion, and this is another limitation of this study.

The results of this study showed that circulating irisin, adiponectin, and RBP-4 levels may be new biomarkers in addition to risk factors such as DM, obesity, and HT. When PSG is not available, these parameters and clinical data can be used to diagnose the disease. This study will help to clarify the formation mechanism of OSAS; we think that it will provide important information in terms of both curative and preventive medicine and will facilitate the recognition of this often-overlooked health problem, even if it is very common. However, more comprehensive studies are required to be able to confirm this issue.

## Figures and Tables

**Figure 1 biomolecules-13-01440-f001:**
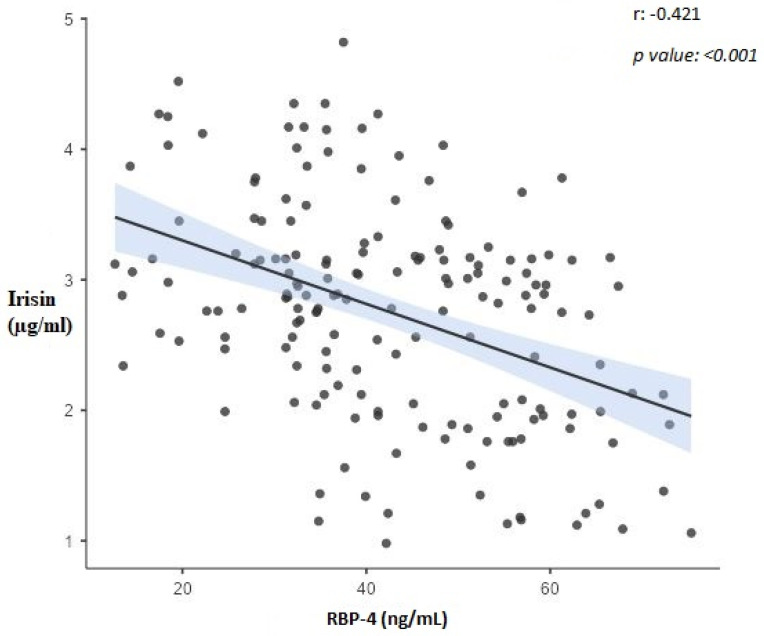
The correlation between Irisin and RBP-4.

**Figure 2 biomolecules-13-01440-f002:**
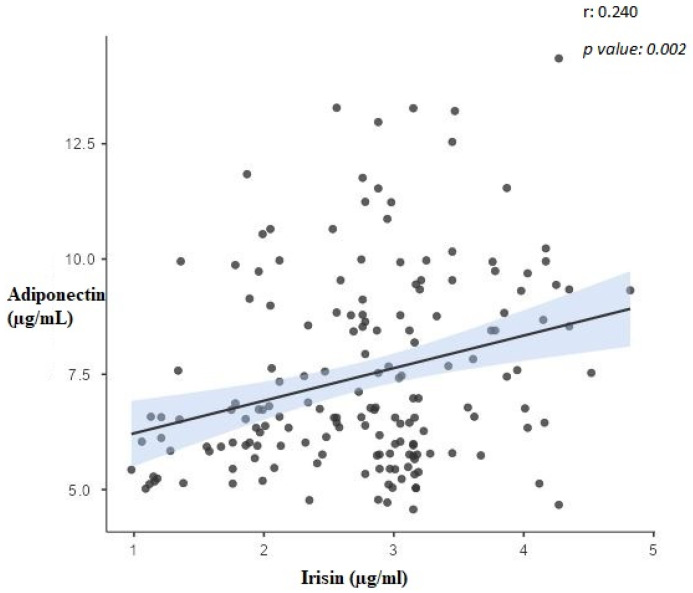
The correlation between adiponectin and irisin.

**Figure 3 biomolecules-13-01440-f003:**
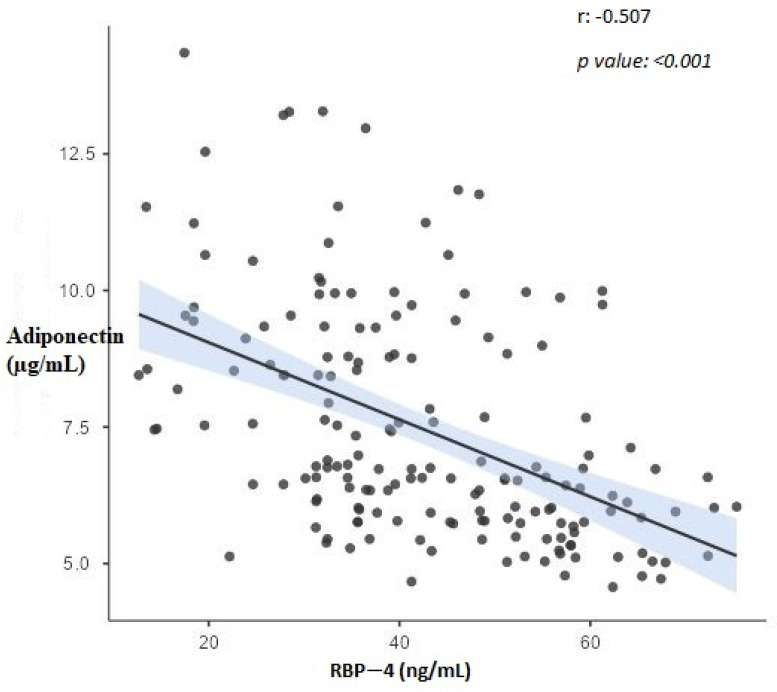
The correlation between adiponectin and RBP-4.

**Table 1 biomolecules-13-01440-t001:** Demographic, sleep recording variables, and laboratory findings of the groups.

Variables	AHI Groups				*p*
	Control AHI < 5 (*n*:46)	Mild OSAS (5 < AHI < 15) (*n*:14)	Moderate OSAS (15 < AHI < 30) (*n*:23)	Severe OSAS (AHI > 30) (*n*:88)	
**Age (Year)**	41.67 ± 11.04 ^a^	49.92 ± 11.25 ^ab^	48.55 ± 13.97 ^ab^	52.57 ± 10.62 ^b^	**<0.001 ***
**Height (m)**	170.13 ± 9.84	165.00 ± 9.58	167.09 ± 10.23	167.93 ± 7.95	0.228 *
**Weight (kg)**	74.61 ± 13.91 ^a^	80.50 ± 8.28 ^ac^	91.13 ± 19.20 ^bc^	93.55 ± 16.37 ^b^	**<0.001 ***
**BMI (** **kg/m^2^)**	25.66 ± 3.59 ^a^	29.76 ± 4.17 ^ac^	32.91 ± 8.11 ^bc^	33.34 ± 6.52 ^bc^	**<0.001 ***
** MOS (%)**	93.00 (90.00–96.75) ^a^	90.00 (85.00–90.00) ^a^	81.00 (75.00–87.00) ^ab^	76.00 (66.25–83.00) ^b^	**<0.001** ¶
**SpO_2_ (%)**	95.50 (92.00–97.00)	94.50 (91.00–95.25)	93.00 (91.00–94.00)	92.00 (90.00–94.00)	0.083 ¶
**Irisin (µg/mL)**	3.20 ± 0.79 ^a^	3.52 ± 0.67 ^a^	2.96 ± 0.23 ^a^	2.34 ± 0.76 ^b^	**<0.001 ***
**RBP-4 (ng/mL)**	28.00 ± 9.58 ^a^	34.64 ± 5.96 ^ab^	42.81 ± 11.72 ^b^	51.24 ± 11.13 ^c^	**<0.001 ***
**Adiponectin (µg/mL)**	8.74 (7.58–10.57) ^a^	6.34 (5.73–7.40) ^b^	5.99 (5.49–7.94) ^b^	6.67 (5.43–8.92) ^b^	**<0.001** ¶
**Gender**	*n* (%)	*n* (%)	*n* (%)	*n*(%)	
** Female**	15 (32.6)	6 (42.9)	7 (30.4)	21 (23.9)	0.431 †
** Male**	31 (67.4)	8 (57.1)	16 (69.6)	67 (76.1)	
**Obesity**					
**No**	22 (47.8)	8 (57.1)	8 (34.8)	28 (31.8)	0.137 †
**Yes**	24 (52.2)	6 (42.9)	15 (65.2)	60 (68.2)	
**DM**					
**No**	46 (100.0) ^a^	13 (100.0) ^ab^	21 (91.3) ^ab^	69 (78.4) ^b^	**0.001 ●**
**Yes**	0 (0.0) ^a^	0 (0.0) ^ab^	2 (8.7) ^ab^	19 (21.6) ^b^	
**HT**					
**No**	45 (97.8) ^a^	10 (76.9) ^b^	11 (47.8) ^b^	49 (56.3) ^b^	**<0.001 ●**
**Yes**	1 (2.2) ^a^	3 (23.1) ^b^	12 (52.2) ^b^	38 (43.7) ^b^	

**AHI**, apnea hypopnea ındex; **MOS,** minimum oxygen saturation; **SpO_2_**: oxygen saturation calculated through pulse oxymeter; **RBP-4,** retinol binding protein; **DM**, diabetes mellitus; **HT**, hypertension. Values are shown as mean ± standard deviation or median (1 st quarter–3 rd quarter) and *n* (%). *p* < 0.05 values are shown in bold. Different superscript letters indicate groups with significant differences. *: One-way ANOVA was applied; ¶: Kruskal–Wallis test was applied; †: Chi-square test was applied; ●: Fishers exact test was applied.

**Table 2 biomolecules-13-01440-t002:** The ANCOVA results of irisin, RBP-4, and adiponectin.

						Interactions between		
	AHI Groups	Age	BMI	DM	HT	Group * DM	Group * HT	Group * HT * DM
	*p* Value							
**Irisin (µg/mL)**	**<0.001**	0.558	0.633	0.965	0.817	0.205	0.497	0.257
** ** **RBP-4 (ng/mL)**	**0.001**	0.971	**0.002**	0.254	0.842	0.539	0.819	0.250
**Adiponectin (µg/mL)**	0.239	0.493	**<0.001**	0.729	0.931	0.574	0.867	0.729

Analysis of Covariance (ANCOVA), a general linear model, was applied. The “*” sign indicates the interaction between the variables.

**Table 3 biomolecules-13-01440-t003:** The comparisons of plasma irisin, RBP-4, and adiponectin levels between comorbidities in cases.

	Obesity	Mean ± Std	*p* Value
**Irisin (µg/mL)**	No	2.87 ± 0.75	0.004 *
	Yes	2.44 ± 0.78	
** ** **RBP-4 (ng/mL)**	No	43.05 ± 10	0.001 *
	Yes	50.43 ± 12.47	
**Adiponectin (µg/mL)**	No	8.78 (6.77–9.92)	<0.001 ¶
	Yes	5.84 (5.39–6.41)	
	**Diabetes Mellitus**		
**Irisin (µg/mL)**	No	2.67 ± 0.78	0.017 *
	Yes	2.21 ± 0.81	
** ** **RBP-4 (ng/mL)**	No	46.3 ± 11.69	0.001 *
	Yes	56.15 ± 10.91	
**Adiponectin (µg/mL)**	No	6.53 (5.76–7.94)	0.009 ¶
	Yes	5.78 (5.29–6.46)	
	**Hypertension**		
**Irisin (µg/mL)**	No	2.62 ± 0.84	0.599 *
	Yes	2.54 ± 0.76	
** ** **RBP-4 (ng/mL)**	No	45.76 ± 12.28	0.02 *
	Yes	50.87 ± 11.45	
**Adiponectin (µg/mL)**	No	6.74 (5.76–8.80)	0.004 ¶
	Yes	5.95 (5.45–6.58)	

*: Independent-sample T test was applied; ¶: Mann–Whitney U test was applied.

**Table 4 biomolecules-13-01440-t004:** Pearson and Spearman correlation coefficients between plasma irisin, adiponectin, RBP-4, and BMI.

Variable	Irisin (µg/mL)	RBP-4 (ng/mL)	Adiponectin (µg/mL)	AHI	BMI (kg/m^2^)
**Irisin (µg/mL)**	1	−0.421 **	0.240 *	−0.834 **	−0.249 *
**RBP-4 (ng/mL)**	−0.421 **	1	−0.507 **	0.473 **	0.535 **
**Adiponectin (µg/mL)**	0.240 *	−0.507 **	1	−0.118	−0.768 **
**AHI**	−0.834 **	0.473 **	−0.118	1	0.205 *

**AHI,** apnea hypopnea index; **BMI,** body mass index; **RBP-4**, retinol binding protein-4. Values with *p* < 0.001 are indicated with ** and *p* < 0.05 are indicated with *. Spearman analysis was used in the correlation of adiponectin with other variables; for all other evaluations, Pearson correlation analysis was used.

**Table 5 biomolecules-13-01440-t005:** Evaluation of ROC analysis for adiponectin, RBP-4, and irisin levels as predictors of OSAS.

Variables	AUC	(CI)	*p*	Cut off Point	Sensitivity	Specificity
**Adiponectin**	0.826	(0.761–0.879)	<0.001	7.42 *	74.4%	84.8%
**RBP-4**	0.893	(0.837–0.935)	<0.001	32.34 ¶	92.0%	67.3%
**Irisin**	0.690	0.615–0.758	<0.001	3.42 *	89.6%	43.5%

**AUC**, area under the curve; **CI**, confidence interval; **RBP-4**, retinol binding protein-4; **BMI**, body mass index. *: lower values; ¶ greater values indicate OSAS.

## Data Availability

Participant-level data are available from the corresponding author.

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
