# Peer review of "Significance of Plasma Irisin, Adiponectin, and Retinol Binding Protein-4 Levels as Biomarkers for Obstructive Sleep Apnea Syndrome Severity"

_biomolecules, 2023, doi:10.3390/biom13101440_

Round 1

Reviewer 1 Report

In this study the authors analyze the levels of irisin, adiponectin and RBP-4 in patients with OSA and healthy subjects, assessing the relation between its levels and OSA severity and the correlations between the biomarkers. Finally, the authors perform a ROC curve and logistic regression testing the capacity of the biomarkers to predict OSA. While this approach could be of great interest, there are many aspects to improve.

General issues

1.      In my opinion the authors should reconsider one of the main conclusions of the paper. In this line, the usefulness of predicting OSA using plasma soluble factors is in some way weak. In contrast, it would be really interesting to test the efficacy of these biomarkers on predicting OSA comorbidities such as DM and HT. Could the authors check if OSA patients with comorbidities had elevated levels of the biomarkers vs. OSA patients without comorbidities?

2.      Also, a major limitation of this study is the selection method for the 46 healthy subjects, who are only gender-paired with the patients. In fact, there are significant differences in terms of age, BMI and the presence of metabolic comorbidities such as DM and hypertension. This means that the differences observed can be due to this cofounding factors more than to the OSA disease itself. Authors try to normalize for this cofounding factor performing logistic regression. I am not a statistic expert, but, I am not sure if these are the correct statistical tests to correct for cofounding factors. Why not to adjust the values with general linear models and plot the adjusted results?

Specific issues.

3.      In the abstract, the authors include some correlations with MOS than then are not shown at the results section.

4.      In the last sentence of the introduction, authors stand that “This study was investigated whether plasma irisin, RBP-4, and adiponectin levels are associated with the severity of OSAS in patients with obesity and type 2 diabetes mellitus (type 2 DM), however the results are focused in patients with OSA, regardless obesity and DM type-

5.      In the introduction the authors mentioned two studies investigating irisin in patients with OSA (refs. 3 and 4) it would be useful to specify something about the results of these studies in the introduction.

6.      In the method section page 3 line 99, healthy individuals […] were confirmed to have OSA please correct this mistake.

7.      In the method section page 3 lines 100-101 authors stand that controls were matched according to age, BMI and gender, which is not in line with study results.

8.      In the method section page 3 line 129 if only EDTA tubes were used, then serum was not obtained, only plasma samples. Please clarify this issue.

9.      Please unless the p-value is <0.0001, authors should specify exact p-value.

10.   Authors did not specify how obesity is defined.

11.   In general, the paper would benefit of some graphs, regarding main results such as irisin, RBP-4 and adiponectin data and some of the principal correlations.

12.   Some paragraphs, of the results section are extremely tedious to read, simply depicting what can be seen at a glance at the tables. Results must be reworded to provide a logical plot thread to the study indicating the meaning of the results and some brief interpretation rather than a simple description of tables.

13.   In this line, the paragraph regarding Table 3 and 4 is for me very confusing, it is not clear whether ROC curve is made for the OSAS, as it stands on the table title, or for the cofounding factors as indicated in the corresponding results text, the same happens with the logistic regressions. In addition, it would be really useful to add some explanation regarding the implications of each test for the study results.

14.   Consider removing duplicated parts of correlations in table 2 as they are confusing.

15.   Consider to display tables 3 and 4 in the same order as they appear in the corresponding results text (they are inversed)

16.   Discussion section page 7 line 256, correlation do not imply causation, the fact that the biomarkers studied correlate do not mean that they are interacting with each other.

17.    The correlations (potential interactions) between adiponectin, RBP-4 and irisin is highlighted as main novelty of this study. So, a paragraph in the discussion section regarding literature on how these factors can interact with each other would be very interesting.

18.   In general, there are some typographical errors. Page 3 line 129 “seperated”, page 3 line 135 “4.9.0%”, page 6 table 3 “cut of” and English language should be revised by a native speaker. 

Author Response

Dear Editor,

First, we would like thank the reviewers for the helpful comments, which led us to conduct appropriate experiments. The revised manuscript has subsequently been rewritten to address these concerns and comments of the reviewers.

We are grateful for your understanding and cooperation in this matter.

Necessary arrangements have been made. If you want it to be edited again, we can have the article edited in the MDPI edit service.

We believe that the manuscript is now suitable for review. We look forward to your reply.

RESPONSE TO REVIEWERS:

Reviewer 1

Comments and Suggestions for Authors

Open Review

( ) I would not like to sign my review report

(x) I would like to sign my review report

Quality of English Language

( ) I am not qualified to assess the quality of English in this paper

( ) English very difficult to understand/incomprehensible

( ) Extensive editing of English language required

( ) Moderate editing of English language required

(x) Minor editing of English language required

( ) English language fine. No issues detected

Yes      Can be improved      Must be improved    Not applicable

Does the introduction provide sufficient background and include all relevant references?

( )        (x)       ( )        ( )

Are all the cited references relevant to the research?

(x)       ( )        ( )        ( )

Is the research design appropriate?

( )        ( )        (x)       ( )

Are the methods adequately described?

( )        ( )        (x)       ( )

Are the results clearly presented?

( )        ( )        (x)       ( )

Are the conclusions supported by the results?

( )        ( )        (x)       ( )

Comments and Suggestions for Authors

In this study the authors analyze the levels of irisin, adiponectin and RBP-4 in patients with OSA and healthy subjects, assessing the relation between its levels and OSA severity and the correlations between the biomarkers. Finally, the authors perform a ROC curve and logistic regression testing the capacity of the biomarkers to predict OSA. While this approach could be of great interest, there are many aspects to improve.

General issues

  1. In my opinion the authors should reconsider one of the main conclusions of the paper. In this line, the usefulness of predicting OSA using plasma soluble factors is in some way weak. In contrast, it would be really interesting to test the efficacy of these biomarkers on predicting OSA comorbidities such as DM and HT. Could the authors check if OSA patients with comorbidities had elevated levels of the biomarkers vs. OSA patients without comorbidities?

Thanks to the reviewer for their valuable comment. A new comparison table and additions to the results have been made.

  1. Also, a major limitation of this study is the selection method for the 46 healthy subjects, who are only gender-paired with the patients. In fact, there are significant differences in terms of age, BMI and the presence of metabolic comorbidities such as DM and hypertension. This means that the differences observed can be due to this cofounding factors more than to the OSA disease itself. Authors try to normalize for this cofounding factor performing logistic regression. I am not a statistic expert, but, I am not sure if these are the correct statistical tests to correct for cofounding factors. Why not to adjust the values with general linear models and plot the adjusted results?

We thank the reviewer for their valuable comments. We agree with the reviewer on this issue. We have created a new table with the results of the generalized linear model and mentioned it in the results.

 Specific issues.

  1. In the abstract, the authors include some correlations with MOS than then are not shown at the results section.

Correlations with MOS are incorrect. It is a secretariat mistake. We have therefore removed it from the abstract.

  1. In the last sentence of the introduction, authors stand that “This study was investigated whether plasma irisin, RBP-4, and adiponectin levels are associated with the severity of OSAS in patients with obesity and type 2 diabetes mellitus (type 2 DM), however the results are focused in patients with OSA, regardless obesity and DM type-

  1. In the introduction the authors mentioned two studies investigating irisin in patients with OSA (refs. 3 and 4) it would be useful to specify something about the results of these studies in the introduction.

Necessary additions have been made.

  1. In the method section page 3 line 99, healthy individuals […] were confirmed to have OSA please correct this mistake.

Corrected as ''.......confirmed to have not OSAS by polysomnography (PSG; AHI <5).''

  1. In the method section page 3 lines 100-101 authors stand that controls were matched according to age, BMI and gender, which is not in line with study results.

Corrected as ‘’Controls were matched according to gender.’’

  1. In the method section page 3 line 129 if only EDTA tubes were used, then serum was not obtained, only plasma samples. Please clarify this issue.

Blood samples were collected, in EDTA containing tubes and anticoagulant-free tubes in the morning after 12-14 hours of fasting. After centrifugation at 2500 x g for 5 min, the plasma and serum seperated at least in 30 minutes.

We have already explained it as "EDTA containing tubes and anticoagulant-free tubes" in the method section. Plasma is obtained from tubes containing EDTA and serum from tubes without anticoagulant.

  1. Please unless the p-value is <0.0001, authors should specify exact p-value.

We thank the reviewer for their comment. We have corrected all p values.

  1. Authors did not specify how obesity is defined.

Necessary additions have been made.

  1. In general, the paper would benefit of some graphs, regarding main results such as irisin, RBP-4 and adiponectin data and some of the principal correlations.

Figures added.

  1. Some paragraphs, of the results section are extremely tedious to read, simply depicting what can be seen at a glance at the tables. Results must be reworded to provide a logical plot thread to the study indicating the meaning of the results and some brief interpretation rather than a simple description of tables.

Necessary additions have been made.

  1. In this line, the paragraph regarding Table 3 and 4 is for me very confusing, it is not clear whether ROC curve is made for the OSAS, as it stands on the table title, or for the cofounding factors as indicated in the corresponding results text, the same happens with the logistic regressions. In addition, it would be really useful to add some explanation regarding the implications of each test for the study results.

We thank the reviewer for their comment. Here we aimed to assess the ability of these markers as predictors for OSAS. Table title corrected. More descriptive edits were made in the results.

  1. Consider removing duplicated parts of correlations in table 2 as they are confusing.

BMI removed from horizontal column

  1. Consider to display tables 3 and 4 in the same order as they appear in the corresponding results text (they are inversed)

Necessary arrangements have been made.

  1. Discussion section page 7 line 256, correlation do not imply causation, the fact that the biomarkers studied correlate do not mean that they are interacting with each other.

''interacting'' corrected as ''relationship''.

  1. The correlations (potential interactions) between adiponectin, RBP-4 and irisin is highlighted as main novelty of this study. So, a paragraph in the discussion section regarding literature on how these factors can interact with each other would be very interesting.

Necessary additions were made according to your suggestions.

 Comments on the Quality of English Language

  1. In general, there are some typographical errors. Page 3 line 129 “seperated”, page 3 line 135 “4.9.0%”, page 6 table 3 “cut of” and English language should be revised by a native speaker.

Necessary corrections have been made.

Reviewer 2 Report

1)    The purpose of this paper wrote in line 83-85. So, authors should write the relationship between the plasma irisin RBP-4 and adiponectin levels, and OSAS patients with obesity and type 2 DM in Discussion.

2)    I do not understand the meaning of p values in Table 1. Are those vs control?

Also, in Table 1: What does it mean of “a” for Age, MOS, Irisin, DM, HT?

               What does it mean of “b” for Age, Adiponectin, DM, HT?

               What does it mean of “c” for BMI, RBP-4?

3)    In Table 2 : What does it mean of AHI/h?

    The correlation coefficient between BMI and AHI is 0.203, without *, from Line215.

4)    Line 154: irisinnàirisin

Author Response

Dear Editor,

First, we would like thank the reviewers for the helpful comments, which led us to conduct appropriate experiments. The revised manuscript has subsequently been rewritten to address these concerns and comments of the reviewers.

We are grateful for your understanding and cooperation in this matter.

Necessary arrangements have been made. If you want it to be edited again, we can have the article edited in the MDPI edit service.

We believe that the manuscript is now suitable for review. We look forward to your reply.

RESPONSE TO REVIEWERS:

Reviewer 2

Open Review

( ) I would not like to sign my review report

(x) I would like to sign my review report

Quality of English Language

(x) I am not qualified to assess the quality of English in this paper

( ) English very difficult to understand/incomprehensible

( ) Extensive editing of English language required

( ) Moderate editing of English language required

( ) Minor editing of English language required

( ) English language fine. No issues detected

Yes      Can be improved      Must be improved    Not applicable

Does the introduction provide sufficient background and include all relevant references?

(x)       ( )        ( )        ( )

Are all the cited references relevant to the research?

(x)       ( )        ( )        ( )

Is the research design appropriate?

( )        (x)       ( )        ( )

Are the methods adequately described?

(x)       ( )        ( )        ( )

Are the results clearly presented?

( )        (x)       ( )        ( )

Are the conclusions supported by the results?

( )        (x)       ( )        ( )

Comments and Suggestions for Authors

1)    The purpose of this paper wrote in line 83-85. So, authors should write the relationship between the plasma irisin RBP-4 and adiponectin levels, and OSAS patients with obesity and type 2 DM in Discussion.

Necessary additions were made according to your suggestions.

2)    I do not understand the meaning of p values in Table 1. Are those vs control?

These results are one-way ANOVA and Kruskall-Wallis results between the 4 groups. For significant results, post-hoc comparisons were made between each group; groups that differed were indicated with different superscript letters.

For example irisin, it is found that a significant difference between four groups. After that we conducted a post-hoc analysis, and it is found that severe OSAS group has lower irisin level compared to other groups.

Also, in Table 1: What does it mean of “a” for Age, MOS, Irisin, DM, HT?

It is explained under the table. Different superscript letters indicate groups with significant differences.

               What does it mean of “b” for Age, Adiponectin, DM, HT?

               It is explained under the table. Different superscript letters indicate groups with significant differences.

               What does it mean of “c” for BMI, RBP-4?

                It is explained under the table. Different superscript letters indicate groups with significant differences.

3)    In Table 2: What does it mean of AHI/h?

            It is a secretary error. AHI/h has been corrected to AHI.

    The correlation coefficient between BMI and AHI is 0.203, without *, from Line215.

    Thank you for your valuable comment. Necessary corrections have been made.

4)    Line 154: irisinnàirisin

            Corrected as irisin.

Round 2

Reviewer 1 Report

Thank you for addressing all the comments. I would only like to ask the authors if their biomarkers are specially elevated in patients with OSA related comorbidites sich as DM. Also, discussion would benefit of an explanation about the potential advantages of diagnosing OSA with their biomarkers vs canonical diagnosis (e.g. polisomnography)

There are still sentences that need to be reworded

Author Response

Dear Editor,

First, we would like thank the reviewers for the helpful comments, which led us to conduct appropriate experiments. The re-revised manuscript has subsequently been rewritten to address these concerns and comments of the reviewers. The article was edited again.

We are grateful for your understanding and cooperation in this matter.

Necessary arrangements have been made. If you want it to be edited again, we can have the article edited in the MDPI edit service.

We believe that the manuscript is now suitable for review. We look forward to your reply.

RESPONSE TO REVIEWERS:

Reviewer 1

Open Review

( ) I would not like to sign my review report

(x) I would like to sign my review report

Quality of English Language

( ) I am not qualified to assess the quality of English in this paper

( ) English very difficult to understand/incomprehensible

( ) Extensive editing of English language required

( ) Moderate editing of English language required

(x) Minor editing of English language required

( ) English language fine. No issues detected

Yes      Can be improved      Must be improved    Not applicable

Does the introduction provide sufficient background and include all relevant references?

(x)       ( )        ( )        ( )

Are all the cited references relevant to the research?

(x)       ( )        ( )        ( )

Is the research design appropriate?

( )        (x)       ( )        ( )

Are the methods adequately described?

(x)       ( )        ( )        ( )

Are the results clearly presented?

( )        (x)       ( )        ( )

Are the conclusions supported by the results?

( )        ( )        (x)       ( )

Comments and Suggestions for Authors

Thank you for addressing all the comments. I would only like to ask the authors if their biomarkers are specially elevated in patients with OSA related comorbidites sich as DM.

Thank you for your valuable comments.

The result has already been written in the following form.

Table 3 shows irisin, RBP-4 and adiponectin levels according to comorbidity groups. Irisin and adiponectin levels were significantly lower and RBP-4 levels were significantly higher in patients with obesity compared to those without obesity. Irisin and adiponectin levels were significantly lower and RBP-4 levels were significantly higher in patients with DM compared to those without DM. RBP-4 levels were signifi-cantly higher and adiponectin levels were significantly lower in patients with hyper-tension compared to those without hypertension, whereas there was no significant difference between irisin levels.

Biomarkers were assessed in patients with OSAS-related comorbidities such as DM HT and obesity. Accordingly, irisin and adiponectin levels decreased and RBP-4 levels increased in patients with obesity and DM. RBP-4 levels were higher and adiponectin levels were lower in patients with HT. These results seen in OSAS patients may be the result of nocturnal hypoxaemia and excessive sympathetic activation in these patients. Indeed, the negative correlation between AHI and adiponectin and irisin and positive correlation with RBP-4 support this argument. Circulating adiponectin, irisin, RBP-4 may be new biomarkers in OSAS patients in addition to major cardiovascular risk factors such as diabetes, obesity and HT.

Also, discussion would benefit of an explanation about the potential advantages of diagnosing OSA with their biomarkers vs canonical diagnosis (e.g. polisomnography).

The gold standard for diagnosis in patients with suspected OSAS is PSG performed in a sleep laboratory (35). AHI is the most commonly used criterion for the diagnosis and severity of OSAS as a result of polysomnographic study. The current study, corre-lation coefficient between RBP-4 and AHI variables shows a statistically significant pos-itive moderate relationship. The correlation coefficient between adiponectin and AHI variables did not show statistically significant relationship. There was a positive corre-lation between AHI and BMI in patient group. Although PSG has been accepted as the gold standard in the diagnosis of OSAS since 1965, it is expensive, time-consuming, and requires special teams and devices (36). Due to the relationship between AHI and cir-culating adiponectin, irisin and RBP-4 levels, they may be preferred in the future be-cause they are cheap, easy and short-term in cases where PSG is not performed. When PSG is not available, these parameters and clinical data can be used to diagnose the disease.

Comments on the Quality of English Language

There are still sentences that need to be reworded

The article was edited again.